# Next Generation of Ovarian Cancer Detection Using Aptamers

**DOI:** 10.3390/ijms24076315

**Published:** 2023-03-28

**Authors:** Rayane da Silva Abreu, Deborah Antunes, Aline dos Santos Moreira, Fabio Passetti, Julia Badaró Mendonça, Natássia Silva de Araújo, Tayanne Felippe Sassaro, Anael Viana Pinto Alberto, Nina Carrossini, Priscila Valverde Fernandes, Mayla Abrahim Costa, Ana Carolina Ramos Guimarães, Wim Maurits Sylvain Degrave, Mariana Caldas Waghabi

**Affiliations:** 1Laboratório de Genômica Funcional e Bioinformática, Instituto Oswaldo Cruz, Fundação Oswaldo Cruz (FIOCRUZ), Rio de Janeiro 21040-900, Brazil; 2Instituto Carlos Chagas, Fundação Oswaldo Cruz (FIOCRUZ), Curitiba 81350-010, Brazil; 3Laboratório de Tecnologia de Pós, Instituto Nacional de Tecnologia, Rio de Janeiro 20081-312, Brazil; 4Divisão de Patologia (DIPAT), Instituto Nacional do Câncer (INCA), Rio de Janeiro 20220-400, Brazil; 5Laboratório de Tecnologia Imunológica, Instituto de Tecnologia em Imunobiológicos, Bio-Manguinhos, Fundação Oswaldo Cruz (FIOCRUZ), Rio de Janeiro 21040-900, Brazil

**Keywords:** ovarian cancer, aptamers, computational modeling

## Abstract

Ovarian cancer is among the seven most common types of cancer in women, being the most fatal gynecological tumor, due to the difficulty of detection in early stages. Aptamers are important tools to improve tumor diagnosis through the recognition of specific molecules produced by tumors. Here, aptamers and their potential targets in ovarian cancer cells were analyzed by in silico approaches. Specific aptamers were selected by the Cell-SELEX method using Caov-3 and OvCar-3 cells. The five most frequent aptamers obtained from the last round of selection were computationally modeled. The potential targets for those aptamers in cells were proposed by analyzing proteomic data available for the Caov-3 and OvCar-3 cell lines. Overexpressed proteins for each cell were characterized as to their three-dimensional model, cell location, and electrostatic potential. As a result, four specific aptamers for ovarian tumors were selected: AptaC2, AptaC4, AptaO1, and AptaO2. Potential targets were identified for each aptamer through Molecular Docking, and the best complexes were AptaC2-FXYD3, AptaC4-ALPP, AptaO1-TSPAN15, and AptaO2-TSPAN15. In addition, AptaC2 and AptaO1 could detect different stages and subtypes of ovarian cancer tissue samples. The application of this technology makes it possible to propose new molecular biomarkers for the differential diagnosis of epithelial ovarian cancer.

## 1. Introduction

Ovarian cancer is the seventh type of cancer with the highest incidence in the female population, and the one with the worst prognosis. Amongst gynecological tumors, it is the most lethal and the third with the highest incidence, surpassed only by cervical and uterine tumors [1]. Although the prevalence of this type of cancer by itself is not expressively high, its lethality reaches significant rates. As an example, ovarian cancer is three times more lethal than breast cancer [2]. This high mortality rate is due to three main factors: silent and asymptomatic tumor growth, late onset of symptoms, and lack of proper population screening, which very often results in the diagnosis of advanced stage disease [1]. Thus, most patients are diagnosed with metastatic disease, which results in less than 30% survival in a five-year timespan. However, when treated early, survival can reach 90% [3].

Currently, the most used biomarker for clinical diagnosis and prognosis in ovarian cancer is based on the serum antigen 125 (CA-125). In late-stage cases of epithelial ovarian cancer, the CA-125 value is elevated by approximately 90%. However, it is only elevated in 50–60% of women with early-stage disease, as well as in several benign conditions, leading to inconclusive diagnosis. Thus, there is an urgent need to identify new methods capable of detecting ovarian tumors in early stages [4,5,6].

In tumor cells, the levels of some cell surface markers may increase, decrease, and undergo modifications, or new markers may appear. These molecular variations can differentiate healthy cells from tumor cells [7,8]. In this sense, the use of aptamers represents a promising approach which could improve specificity in tumor diagnosis. Aptamers are small, single-stranded synthetic oligonucleotides (DNA or RNA, with 40 to 80 nucleotides) that bind with high specificity to a molecular target and are, therefore, strong candidates for diagnostic tests and therapy [9]. Aptamers are selected through an in vitro process called “Systematic Evolution of Ligands through Exponential Enrichment” (SELEX) which can result in the recognition of a wide variety of target molecules, from small structures to macromolecules [10,11]. Aside from the classic SELEX, cell-based SELEX (Cell-SELEX) could also be applied as a method of selection of aptamers in whole cells, especially for the detection of surface proteins that have increased expression in tumor cells [12]. The use of Cell-SELEX has advantages, as it is based on the selection of aptamers against molecules of living cells that maintain their native conformations and, in this way, preserve their biological functions.

Despite the advantages described above, the Cell-SELEX method does not give clues regarding the aptamer target in the cell. Thus, we used in silico approaches to uncover information about possible tumor targets. Computational tools that allow the prediction of three-dimensional structures of both nucleic acids and target proteins [13,14] could provide essential information for molecular docking simulations. A bioinformatics methodology was proposed for aptamer–target complex validation in three stages, involving: (1) the prediction and (2) modeling of the aptamer nucleic acid structure, followed by (3) molecular docking between the aptamer and the potential protein target in an attempt to reproduce the experimental aptamer–protein interactions [15].

However, the construction of three-dimensional models of DNA aptamers remains a challenge, especially due to the lack of algorithms for the three-dimensional modeling of DNA molecules, considering that most of the available programs are dedicated to the modeling of RNA molecules. One of the most used protocols to solve this issue was developed by Jeddi et al. (2017), which consists of a workflow to predict the modeling of DNA aptamers, based on four steps: (1) prediction of the secondary structure from the ssDNA sequence; (2) construction of ssRNA equivalent models; (3) transformation of ssRNA models into three-dimensional ssDNA structures; and (4) refinement of modeled three-dimensional ssDNA structures [15].

We sought to enhance the selection of aptamers with potential use in the clinical diagnosis of ovarian cancer using computational tools in a complementary way, favoring a better identification of more stable aptamers and their respective targets. The identification of specific aptamers, including those with a potential for detection of metastasis, can help to reduce the gaps in clinical practice regarding ovarian cancer diagnosis. As there are still no methods with reliable sensitivity and specificity for the early diagnosis of epithelial ovarian cancer, aptamers represent an important tool to improve the specificity for the diagnosis, leading to more accurate and faster disease detection, with a consequent increase in the life expectancy of patients.

## 2. Results

### 2.1. Selection and Identification of Tumor-Specific Aptamers for the Caov-3 and OvCar-3 Cell Lines

Two ovarian tumor cell lines were chosen for the selection of aptamers: the ovarian serous adenocarcinoma cell line Caov-3, and the high-grade serous ovarian adenocarcinoma cell line OvCar3, obtained from malignant ascites. In order to identify aptamers that specifically bind to ovarian cancer cells, the non-tumoral ovarian cell line Iose-144 was used for counterselection. The Cell-SELEX procedure is described briefly below.

To start the selection process, 10 umoles of a naive ssDNA aptamers library was enriched by binding to Caov-3 and OvCar-3 cell monolayers, separately. Sequences which presented binding to general ovarian cell surface markers were removed by incubating the enriched pools with Iose-144 cells (rounds 2, 4, 6, 8, 10, and 12 for Caov-3 selection and rounds 2, 4, 6, 8, 10, 12, 14, and 15 for OvCar-3 selection). The eluted pool for each round of Cell-SELEX was amplified through PCR, after which the ssDNA pools of interest were recovered and monitored for enrichment towards ovarian tumor cells by flow cytometry. The sequences binding to Iose-144 cells were successfully removed by counter selection during the Cell-SELEX rounds, while the enrichment towards the target cell line was maintained. After 12 (Caov-3) and 15 (OvCar-3) rounds of Cell-SELEX, two enriched pools that specifically bound to the respective ovarian tumor cell lines, but at most marginally to Iose-144 cells, were obtained. The five final pools from each cell line selection were chosen and submitted for Next-generation sequencing (NGS): rounds 8–12 for Caov-3 and 12–15 for OvCar-3, in order to identify enriched aptamer sequences after the Cell-SELEX procedure.

Aptamer sequences were aligned into families according to sequence homology. The number of homologues was compared across the sequenced pools from each round of selection in order to validate their enrichment using bio-informatics tools. The five most frequent sequences for each cell line (Caov-3 and OvCar-3), showing consistent enrichment throughout the pools (Figure 1A,B, respectively), were selected as the most promising aptamer candidates. These were then synthesized and tested for binding to the ovarian cancer cell lines. The aptamers AptaC2, selected for Caov-3, and AptaO1, selected for OvCar-3, presented a continuous enrichment throughout the selection process, suggesting that they are more stable sequences or aptamers targeted to highly expressed cell targets.

Flow cytometry analysis confirmed the binding specificity of the selected aptamers for the ovarian tumor cells (Figure 2A–D). Interestingly, aptamers selected for Caov-3 cells were able to recognize OvCar-3 cells (Figure 2B), and those selected for OvCar-3 also recognized Caov-3 cells (Figure 2D) with different intensities, as demonstrated by the median fluorescence intensity (MFI) (Figure 2E,F).

Next, we aimed to characterize possible interactions between aptamers and their potential targets in the tumor cells, and to do so, reliable structures needed to be obtained for both aptamer and protein. For this purpose, in silico approaches were performed to characterize the 3D structure from the five selected aptamers from both cell lines Caov-3 and OvCar-3. In order to analyze possible aptamer targets, Caov-3 and OvCar-3 cell line proteomic data from the literature were analyzed [16,17]. The sequential steps from aptamer analysis and potential target protein analysis are described in a pipeline (Figure 3).

### 2.2. Structural Characterization of Selected Aptamers for Caov-3 and OvCar-3 Cells

As a first step, we performed the prediction of secondary structures from the selected aptamers, using as an input the ssDNA sequences of each of the ten aptamers (five from Caov-3 and five from OvCar-3). The secondary structures of the ssDNA molecules were predicted using the mfold web server (http://mfold.rna.albany.edu/?=mfold, accessed on 4 March 2022) and NUPACK web server (http://www.nupack.org, accessed on 2 March 2022), based on free energy minimization techniques. In mfold, all possible secondary structures are calculated based on Watson–Crick base pairing, and the most thermodynamically stable structures are then selected. This step also provides the minimum free energy of each aptamer fold. The ssDNA sequences were selected considering the lower ΔG value, indicating more stable structures, as well as the similarity between predicted structures from both mfold and NUPACK.

Among all five aptamers selected for Caov3, AptaC4 presented very similar predicted structures with both applications, mFold and NUPACK. The predicted structures for the Caov-3 aptamers showed greater internal pairing between the sequences, with a large number of hairpins (Appendix A). Among all five aptamers selected for OvCar-3, AptaO1 showed the most similar structure prediction between the two applications used for analysis. In general, after comparing results obtained by both programs, we observed that the predicted structures showed only few regions with internal nucleotide pairing, except in the cases of AptaO1 and AptaO2. Most structures have large loops (Appendix A). The structures with lower free energy (ΔG) were considered for each model, since one expects the most spontaneous and stable aptamer folding when lower dissipated energy is observed. Appendix A (Caov3) and Appendix A (OvCar-3) describe the ΔG values obtained for the predicted structure of each aptamer, provided by both applications. All ΔG values obtained for the secondary structure of each analyzed aptamer sequence were negative values, indicating that these structural conformations are potentially spontaneous (Appendix A). Again, AptaC4 (Appendix A) and AptaO1 (Appendix A) were the most stable structures, showing the same calculated ΔG values by mFold and NUPACK predictions. Thus, the aptamers AptaC2 and AptaC4, identified for Caov-3 cell line, and AptaO1 and AptaO2, identified for Ovcar-3 cell line, were selected for further analysis. (Figure 4).

There is a lack in the literature of in silico protocols and methodologies capable of performing three-dimensional modeling of ssDNA aptamers. Herein, we refine a computational method to predict ssDNA 3D structures based on the methodology developed by Jeddi and Saiz (2017). We first used as an input in the RNA Composer the secondary consensus structures, which were obtained from the prediction of both applications, mfold and NUPACK. A 3D RNA prediction was obtained based on 2D ssDNA sequences. Next, using the 3DNA server, each of the 3D RNA structures were transformed into 3D ssDNA by the exchange of ribose sugars to deoxyribose and uracil bases to thymine, in an automated way [15].

Representative 3D structures for AptaC2, AptaC4, AptaO1, and AptaO2, predicted from their respective secondary structure information, are shown (Figure 4). However, ssDNA molecules present very flexible structures, especially in the case of aptamers with little base pairing in their conformational structures, which is particularly observed for the predicted models in this analysis. Thus, molecular dynamics simulations could help to identify the most stable three-dimensional conformation and improve the predicted models.

### 2.3. Characterization of the Three-Dimensional Structure Stability of Selected Aptamers for CaoV-3 and OvCar-3 Cells by Molecular Dynamics

For the selected aptamers, AptaC2, AptaC4, AptaO1, and AptaO2 molecular dynamics simulations were to evaluate the stability of the predicted 3D structures. We have analyzed the aptamer’s conformational behavior and verified structural stability using root-mean-square deviation (RMSD) and radius of gyration (Rg), taking their respective final configurations from the equilibration stage as a reference. During the production stage’s transitory regime, all aptamers showed high RMSD values above 10 Å, and the relaxation time necessary to achieve stability for the aptamers varied from 200 ns to 300 ns (Figure 5A). In order to understand what contributed to the high disruption of the initial configuration of aptamers, we evaluated the compaction of the structures through Rg calculations (Figure 5B).

All aptamers decreased their Rg from about 25 Å to 22.5 Å. The exception was the AptaC4, which maintained an Rg of approximately 30 Å. Additionally, we performed an RMSD-based clustering analysis (cut-off = 5 Å) to identify the relevant conformational changes in the structures over the simulations (Appendix A). For each system, the central structure of the largest cluster was taken and compared with its respective final configuration from the equilibration stage. These analyses demonstrate that the high RMSD values are primarily attributable to the 5’ and 3’ ends of the ssDNA, which were compacted and rearranged throughout the Molecular Dynamics (MD) simulations (Figure 5).

Hydrogen bonds and aromatic stacking impact the structure and dynamics of nucleic acids. Thus, we assessed the evolution of the secondary structure of aptamers along the trajectories to identify variation in the annotation of a set of structures (Figure 6). These analyses showed that, despite the notable change at the three-dimensional level, the base-pairing of the secondary structure predictions were maintained throughout the simulations. Sugar/Watson–Crick interactions were the most common and stable pairings, with an occupancy (percentage of simulation time) of over 95%.

### 2.4. Selection of Potential Aptamer Targets in Caov-3 and OvCar-3 Cell Lines

Aiming to identify potential protein targets that could be recognized and bound by the aptamers in tumor cells, proteomic data from the Caov-3 and OvCar-3 cell lines provided by Coscia et al. (2016) and Faça et al. (2008) were analyzed. First, in order to identify proteins differentially expressed in Caov-3 cells or in Ovcar-3 cells, a 1.5× fold change was considered as an initial filter, where selected proteins are overexpressed in Caov-3 and OvCar-3 cells compared to the non-tumor control cell lines. From this first filter, 486 (Caov-3) and 512 (OvCar-3) proteins were identified as overexpressed according to data obtained in Coscia et al. (2016). Then, after confirming the overexpression in a second proteomic study provided by Faça et al. (2008), a final list was obtained. This list contained 93 proteins for Caov-3 and 96 proteins for OvCar-3 considered as possible targets for aptamers selected in Caov-3 and OvCar-3 cells, respectively, which were ranked according to their expression levels [16,17].

Since aptamers from the Cell-Selex screening are expected to recognize cell surface targets, subsequently, the 93 proteins from the Caov-3 list and 96 proteins from the OvCar-3 list were analyzed as to their cellular localization, preferentially in the plasma membrane, according to information obtained from UniProt and PSORT II. As a result, 26 proteins from Caov-3 (Appendix A) and 24 proteins from OvCar-3 (Appendix A) were identified with cell localization reported in plasma membranes containing transmembrane domains (Appendix A).

As molecular docking is based on reliable predicted 3D structures, a search for 3D predicted structures of the selected proteins was performed using the PDB and AlphaFold protein structure database. Thus, we obtained 10 proteins from Caov-3 and 7 proteins from OvCar-3 lists with predicted 3D structures, containing well-resolved extracellular portions.

### 2.5. Characterization of Spatial Orientation and Electrostatic Potential of Membrane Proteins Selected for Caov-3 and OvCar-3 Cells

In order to identify which proteins, among those listed as possible targets for aptamers in ovary cancer cells, present residues in the extracellular portion, the proteins’ spatial orientation in the lipid bilayer was characterized using the Orientations of Proteins and Membranes (OPM) server. OPM provides spatial arrangements of membrane proteins with respect to the hydrocarbon core of the lipid bilayer. Red and blue pseudo atoms mark the hydrophobic boundaries of the lipid bilayer, while the red portion indicates the extracellular region, and the blue portion indicates the intracellular region.

Furthermore, to determine the best candidates as potential targets for aptamer binding in tumor cells, the molecular surface electrostatic potential of the membrane proteins selected for Caov-3 and OvCar-3 was also evaluated with the Adaptive Poisson-Boltzmann Solver (APBS) server to determine the amino acids charge. Electrostatics is an important driving force in nucleic acid binding, so the selection of proteins that have a molecular surface with positively charged amino acids, favoring the binding to negatively charged nucleic acid sequences, is a relevant step in the evaluation of possible interaction targets. The results are represented by the color variation observed in the molecular surface of the protein, such as blue (indicating regions of positive charge) and red (indicating regions of negative charge), and the intensity of the color is proportional to the intensity of the charges in the molecular region.

Due to their structural characteristics, from the selected proteins of the Caov-3 (10 proteins) and OvCar-3 (7 proteins) lists, six proteins from each cell line were considered as possible targets for aptamers. The proteins presenting spatial orientation with large portions of their structure oriented towards the extracellular region of the lipid bilayer with positively charged regions (represented by the intensity of the blue color), which indicates a positive electrostatic potential in these areas, are shown for Caov-3 (Figure 7) and OvCar-3 (Figure 8) cell lines. Then, these 12 proteins were chosen for the next step of molecular docking.

### 2.6. Characterization of Protein-Aptamer Complexes by Molecular Docking

Molecular docking analyses were performed by the HADDOCK web server in order to identify the best protein–aptamer binding complexes, and consequently indicate a possible protein target with the greatest potential for aptamer recognition. The central structure of the largest cluster of the molecular dynamics’ simulation was taken for molecular docking analyses.

As an input, each of the six selected proteins from Caov-3 was docked with AptaC2 and AptaC4, separately, resulting in 12 molecular dockings for Caov-3 cells. In addition, each of the six selected proteins from OvCar-3 was docked with AptaO1 and AptaO2, separately, resulting in 12 molecular dockings for OvCar-3 cells. After molecular docking procedures, to choose the best complexes of protein-aptamer, the following parameters were considered: docking score, cluster size, z-score, and RMSD. The score values obtained for each of the complexes formed after molecular docking are described for Caov-3 cell line (Table 1) and OvCar-3 cell line (Table 2). According to the score value and the binding position, the complexes AptaC2-FXYD3, AptaC4-ALPP, AptaO1-TSPAN15, and AptaO2-TSPAN15 were selected as the best complexes for each aptamer (Figure 9).

The three potential targets FXYD3, ALPP, and TSPAN15 were further explored regarding their expression profile in non-tumoral (normal), primary site tumor, and metastatic tumor tissues, using GeneChip data from TNMplot database. Corroborating the results obtained from the proteomic data analysis, the TNMplot database showed an increased expression of FXYD3 in tumor and metastatic tissue samples as compared to normal samples (Figure 10A). ALPP expression was higher in primary site tumor samples when compared to normal and metastatic samples (Figure 10B) while TSPAN15 was upregulated in primary site tumor and metastatic samples as compared to normal samples (Figure 10C).

The PPI networks, through STRING, showed that the FXYD3 significantly interacted with ATP1B2, ATP1B1, GPRC5A, ATP1A1, ATP1A2, ATP1A3, and ATP1B3, along with proteins from the FXYD family (FXYD7, FXYD2, FXYD5), all involved in the modulation of ion channels (Figure 10D). TSPAN15 interacted with ITGBB1, APP, PTGFRN, ADAM10, and some members of the Tetraspanin family (TSPAN5, TSPAN14, TSPAN13, and TSPAN32) which were mainly involved in cell adhesion, motility, activation, and proliferation (Figure 10E).

Based on the docking parameters and expression levels of the potential targets, the AptaC2-FXYD3, AptaO1-TSPAN15, and AptaO2-TSPAN15 followed MD simulations on the docking binding poses in order to assess the stability of the complexes.

### 2.7. Characterization of Protein-Aptamer Complex Stability by Molecular Dynamics

A well-conducted molecular dynamics simulation might be useful for discriminating between correct molecular docking poses and false positives. Therefore, MD trajectories were run, considering membrane-embedded proteins FXYD3 and TSPAN15, and processed in order to evaluate the stability and energy of aptamers using H-bonds and MM/GBSA analysis.

Using the MM/GBSA method, we calculated the absolute free binding energy (∆G_bind_) in order to quantify the intensity of the interaction between proteins and aptamers. Overall, the ∆G_bind_ values of all evaluated systems were negative, indicating spontaneity in the recognition process (Table 3). When analyzing the ∆G_bind_ decomposition into individual energies using the MM/GBSA method, we noted that the van der Waals contribution (∆E_vdw_) increased the binding affinity in all systems. However, the electrostatic energy term (∆G_ele+egb_) was primarily responsible for decreasing the binding affinity. TSPAN15 demonstrated a greater affinity for AptaO1 (−27.39 kcal/mol) than AptaO2 (−15.27 kcal/mol), mainly because its electrostatic energy term (∆G_ele+egb_) displayed a lower value for AptaO1.

The MM/GBSA method also supplied the decomposition of the binding energy for each residue. From these values, we examine how the residues from protein and aptamer could contribute to the total binding energy.

For FXYD3, the region between residues 26 and 38 contributed to the interaction energy with AptaC2, with values ranging from −6.74 kcal/mol for Phe-30 to −1.25 kcal/mol for Ser-28 (Figure 11A). In AptaC2, nucleotides 55 to 57 contributed the most, with DT55 contributing −2.30 kcal/mol and DG56 contributing −0.66 kcal/mol (Figure 11B). Simulations indicated that protein could form extensive hydrogen bonds with the aptamer. Notably, Phe-30, which possesses the lowest interaction energy, formed H-bonds with DT55 at an occupancy rate of 89% (Figure 12A). In addition, we emphasize Gln-38, which has an H-bond occupancy of 66% and an energy of −3.37 kcal/mol.

Decomposition of the binding energy of TSPAN15 revealed that most interactions with AptaO1 and AptaO2 aptamers involve the same residues (Figure 11C,E). The distinction pertains to the magnitude of the interaction’s intensity, which was stronger when complexed with AptaO1. As a result, there are significant differences between the systems in terms of H-bond analysis. When complexed with AptaO1, TSPAN15 exhibited an H-bond between Arg-161 and DA8 with a 100% occupancy (Figure 12B) and interaction energy of -10.82 kcal/mol. In AptaO2, the same Arg-161 was incapable of forming H-bonds, and exhibited a weaker interaction energy of -4.13 kcal/mol. With occupancy values greater than 75% in AptaO2 (Figure 12C), Arg-217, which has an energy of -11.78 kcal/mol, interacts with DA6 and DA7. This same amino acid in AptaO1 exhibits a weaker interaction of −4.25 kcal/mol and an H-bond occupancy value of less than 40% (Figure 12B).

### 2.8. AptaC2 and AptaO1 Detect Human Ovarian Tumors from Different Subtypes

After characterizing the most stable aptamers with potential use for ovarian cancer diagnosis, we next aimed to validate its functionality and specificity in tissue samples from patients with ovarian tumors. In this purpose, tissue microarrays were obtained in order to test aptamers’ specificity. Tissue microarrays are paraffin blocks produced by extracting cylindrical tissue cores from different paraffin donor blocks and re-embedding these into a single block, called a microarray. Using this technique, up to 100 tissue samples were arrayed for aptamer binding in a single paraffin block. Human ovarian tumor and non-tumor samples were used to validate the binding capacity and specificity of the selected aptamers. From a cohort of 100 ovarian tissue samples, we analyzed the distribution as follows: 3 ovary non-tumoral tissues, 17 adjacent non-tumoral ovary tissues, 18 benign samples (adenoma), 7 borderline, 10 metastatic carcinoma, 5 invasive urothelial carcinoma, 10 endometrioid adenocarcinoma, 3 mucinous adenocarcinoma, and 27 cases of serous carcinoma. This arrangement makes it possible to assess a high spectrum of ovarian tumor disease.

AptaC2 and AptaO1 were able to specifically recognize malignant ovarian tumor tissue, since it showed no or low intensity of labeling in non-tumoral, benign, and borderline samples (Figure 13A–C). Interestingly, AptaC2, which was selected for Caov-3 cells from primary site tumors, did not detect metastatic samples. On the other hand, AptaO1, which was selected for OvCar-3 cells from metastatic ascites, showed intense labeling in metastatic tumor samples (Figure 13D). In addition, even though it was selected for serous epithelial ovarian tumor cells, AptaC2 and AptaO1 also detected other tumor subtypes, showing intense staining in endometrioid and mucinous ovarian tumor samples (Figure 13E,F). Finally, AptaO1 presented a notable difference in the staining intensity between both serous types: an intense staining was observed in samples of high grade serous type, and a lower staining in the low grade serous type, indicating that this aptamer, AptaO1, could be a marker of ovarian tumor which is capable of differentiating the serous types grades (Figure 13G,H).

## 3. Discussion

Ovarian cancer is the most aggressive of the gynecological tumors. The high lethality associated with ovarian tumors is directly related to the late diagnosis of this type of cancer. To date, there have been no methods with reliable sensitivity and specificity for early epithelial ovarian cancer diagnosis. Thus, aptamers represent as an important tool for improving the specificity of the diagnosis of ovarian cancer, especially because they are molecules capable of binding to a target with high specificity. Aptamers could also be explored for use in imaging exams or even in the identification of tumor biomarkers. In the present study, an aptamer capable of specifically recognizing ovarian tumor samples was identified. In addition, through computational approaches, the three-dimensional structure and stability of this aptamer was characterized, as well as its possible binding targets in tumor cells.

All selected aptamers showed cross recognition between the Caov-3 and OvCar-3 cell lines, despite being selected separately, which could indicate that these aptamers share (elements of) the same targets in tumor cells. On the other hand, they did not recognize Iose-144, the non-tumor control cell line, demonstrating that they are specific for tumor cells. Caov-3 are human adherent epithelial cells of primary site origin [18], while OvCar-3 are adherent cells derived from metastases in malignant ascites of patients with progressive ovarian adenocarcinoma [19]. Iose-144 cells from normal human cell lines of ovarian surface epithelial cells (OSE) have been immortalized (IOSE) by inducing overexpression of the ZNF217 gene [20]. Both tumor cell lines used here are classified histologically as representatives of epithelial-type ovarian tumors; within this classification, they are described as belonging to the group of serous tumors. This is the most frequent, accounting for 80% of cases, and still the most aggressive among ovarian tumor types [21].

There is a growing interest in applications of computational methodologies for studying, modeling, and refinement of aptamers, helping to optimize the identification of the most promising aptamers in a given selection [22,23]. The knowledge about in silico methodologies to characterize the aptamer structure is of great relevance, since it can guide aptamer structural optimization and stabilization experiments. It can also be applied to Molecular Docking tests to identify the best complexes formed by the binding of aptamers with their respective cellular targets, as already demonstrated by Rabal et al. (2016) when performing Molecular Docking assays for aptamer-proteins, and evaluating the interactions involved in this complex [14]. In addition, bioinformatic computational methodologies, such as docking and molecular dynamics, have been proposed as an alternative strategy to SELEX, the traditional method to select aptamers for a specific protein [22]. The development of new methodologies which contribute to the choice of the best protein–aptamer complex identification is especially important for aptamers selected by Cell-SELEX methodology since, in this case, the aptamer target in the cell is unknown.

The modeling of these structures is essential for identifying sequence motifs that favor binding to cellular protein targets, since sequence motifs are regions with recurrent pattern signatures that indicate a protein-binding site [24]. Therefore, we performed analysis with two different applications to compare the fidelity of the prediction of secondary structures of aptamers: mFold and NUPACK. Both are methods based on free energy for secondary structure prediction. The mFold server provides multiple structures considering the lowest free energy through free energies parameters used from the laboratory of John Santa Lucia Jr., considering the application of the thermodynamics nearest-neighbor (NN) model to nucleic acids (1998) [25]. The NUPACK server predicts secondary structures, considering pseudoknotted, and the free energy is calculated using nearest-neighbor empirical parameters in Na+ and Mg++ concentrations [26]. As they use different prediction algorithms, some divergences in the resulting structures provided are expected. However, the fact that AptaO1 and AptaC4 obtain identical results in terms of structures and free energy values, from different prediction methods, allows for confidence in the conformation modeling of these aptamers.

All the free energy values obtained for the secondary structure of each analyzed aptamer sequence were negative values, indicating that these structural conformations are spontaneous. It has already been described that free energy values in aptamers could differ depending on their sequence and thermodynamic profile [27]. However, there are data on aptamers with potential clinical use with a free energy value from −0.61 kcal/mol, such as the PF1 aptamer, a DNA aptamer selected against HIV-1 reverse transcriptase [28].

A better structural characterization of aptamers’ potential target is also needed for determination of the most promising protein–aptamer complexes. While aptamers can be selected against a variety of structural targets, surface protein targets are attractive, as they are more accessible to structural and functional studies. A characterization of the extracellular region of selected proteins for the two cells of interest in the study was carried out by identifying the most promising targets for interaction with aptamers. Information on the topology of membrane proteins allows us to identify the orientation in which the C and N terminals are inserted in the membrane in relation to the extracellular and intracellular regions [29]. According to the classification obtained by PSORT, the membrane proteins selected for OvCar-3 presented these four types of topologies: 1a (a cleavable signal segment and a transmembrane segment), 2 (N-terminal portion inserted in the cytoplasmic region), 3a (multiple regions transmembrane and N-terminal portion inserted on the cytosolic side), and Nt (N-tail topology—contains a non-cleavable signal peptide and a transmembrane segment near the C-terminal portion). The membrane proteins identified for Caov-3 presented topologies of the types 1a, 2, Nt, 3a, and 3b (multiple transmembrane regions and N-terminal portions inserted in the extracellular side). Proteins with type 1a membrane topology are very rare, representing <1%. On the other hand, proteins with type 3a membrane topology are mostly plasma membrane proteins, representing 70% of this group, whereas proteins with type 3b membrane topology are not integral proteins of the membrane [30].

The evaluation of the electrostatic potential of the selected transmembrane proteins allows for the determination of molecules that may have a greater binding affinity with the aptamers, through the assessment of the charge on the molecular surface of each protein. Proteins with positive electrostatic potential are expected to have a higher binding affinity with ssDNA aptamers, due to the negative charge of these molecules. Information about the molecular surface and the electrostatic potential of proteins and DNA has already been used to predict regions of interactions between the two ligands. Tsuchiya et al. (2004), developed statistical evaluation functions considering molecular surface shape and electrostatic potential, and have achieved high success rates [31]. Jones et al. (2003) developed a method of predicting DNA binding sites from residue spots on the proteins’ surfaces, indicating regions with higher positive electrostatic score, while excluding residues with negative electrostatic score. Here, these were considered to be possible aptamer targets: the proteins presenting predominantly positively charged surface, or those with a neutral charge containing positive charge regions [32].

The analysis of molecular docking identified the proteins FXYD3 and ALPP as the most promising targets for aptamers in Caov-3, and for the aptamers selected for OvCar-3, the TSPAN15 protein appeared as being the best target for both aptamers AptaO1 and AptaO2. As a potential target for AptaC2, the protein FXYD3 is a regulator of the activity of sodium/potassium-transporting ATPase (NKA), localized in the cytoplasmic membrane of epithelial cells, and is described as overexpressed in breast cancer, especially in primary tumors and in the ER+ subtype, urothelial carcinoma, and endometrial cancer [33,34,35,36,37]. The potential target for AptaC4 in Caov-3, ALPP, is a surface protein in epithelial cells, and is overexpressed in pancreatic carcinoma [38,39]. TSPAN15, the common target for both aptamers selected for OvCar-3 cells, is a transmembrane protein that is related to esophageal carcinoma, oral carcinoma, liver hepatocellular carcinoma (LIHC), and hepatocellular carcinoma (HCC). Interestingly, for all tumor types, TSPAN15 is described as especially associated with metastasis, indicating that cells overexpressing this protein have increased proliferative, migratory, and invasive capabilities [40,41,42,43]. This information gives robustness to the selection method used here to identify the potential targets for the aptamers. Thus, FXYD3 and ALPP could be used as biomarkers for primary sites of ovarian tumors and TSPAN15 as a biomarker for ovarian cancer metastasis. However, it is essential to emphasize that these targets should be validated by bench techniques, such as surface plasmon resonance (SPR) analysis, which tests the affinity between aptamer-individual recombinant protein complexes; electrophoretic mobility shift assays (EMSA), associated with DNA pull-down assays; or by the use of si/shRNA to knock down these proteins.

The potential use of the selected aptamers for the diagnosis of ovarian cancer was tested in a tissue microarray, consisting, in a broad range, of clinical samples in a single block. With this approach, we could confirm the specificity of the aptamers AptaC2 and AptaO1 for ovarian tumor tissue samples and, moreover, indicate its potential use to detect primary sites tumors and metastatic ones. We could, therefore, propose the combination of both aptamers to increase the detection of ovarian tumors from all subtypes, even those in early-stage disease.

In view of the present data, we hope to contribute to improving the diagnosis of epithelial ovarian cancer, which would diminish the risk of premature death for thousands of women worldwide. We highlight the potential use of the identified proteins FXYD3, ALPP, and TSPAN15 as new molecular biomarkers for ovarian cancer, and the use of the set of aptamers comprising AptaC2, AptaC4, AptaO1, and AptaO2, which should be exploited for clinical applications both in ovarian cancer diagnosis and in targeted cancer therapy. Extrapolating the diagnostic aspect to the specificity of aptamers to tumor cells, avoiding undesirable binding to non-tumoral cells, may offer more efficacious cancer treatment regimens than those in current clinical practices.

## 4. Materials and Methods

### 4.1. Cell Lines

OvCar-3 cells are adherent cells, derived from malignant ascites metastasis of patients with progressive ovarian adenocarcinoma [19], and, therefore, considered to have greater metastatic potential and were used as metastatic cells in the present study. Caov-3 ovarian tumor cells are adherent human epithelial cells of primary site origin, of the high-grade serous type [18], and are therefore considered to have lower metastatic potential. These were used as non-metastatic cells in the present study. Iose-144 cells are human cell lineage derived from ovarian surface epithelial cells (OSE), which were immortalized (IOSE) through the induction of overexpression of the ZNF217 gene [20] and were kindly donated by Dr. Clara Salamanca and used as non-tumor ovarian cell lines.

### 4.2. Cell-SELEX

Non-metastatic (CaoV3) and metastatic (OvCar-3) ovarian tumor cell lines and their respective non-tumor control cells (IOSE-144) were used. Aptamers specific for the OvCar-3 tumor cell lines were previously selected by our group using the Cell-SELEX method after 16 rounds of selection [44]. Successive rounds of Cell-SELEX resulted in the selection of aptamers specifically recognized by metastatic and non-metastatic tumor cells. A library of N30 oligonucleotide DNA aptamers (5 nmol) was initially incubated with the tumor cells for screening and selection of aptamers that bind to molecules present on the cell surface. The bound aptamers were then amplified in a PCR reaction. After 12 rounds (Caov-3 cell line) and 16 rounds (OvCar-3 cell line) of selection and amplification, our experiment resulted in the exponential enrichment of specific aptamers that were submitted to sequencing.

### 4.3. Identification and Analysis of Selected Aptamer Sequences for Caov-3 and OvCar-3 Cells

The identification of aptamer sequences for non-metastatic and metastatic tumors were generated by the next-generation sequencing (NGS) methodologies, MiSeq (Illumina, San Diego, CA, USA) on the NGS Plataform of the Rede de Plataformas Tecnológicas Fiocruz, at the Laboratório de Genômica Funcional e Bioinformática (IOC/Fiocruz, Rio de Janeiro, Brazil). The quality of generated sequences was analyzed for the last five rounds (R8, R9, R10, R11, and R12 for CaoV3 cells and R12, R13, R14, and R15 for OvCar3 cells) using the FastQC tool [45]. The results of the FastQC analyzes were visualized in HTML format files containing sequences’ basic statistics, such as the number and size of generated reads, the distribution of quality values for each one of the bases, and the GC content. Nextera XT adapters (Illumina, San Diego, CA, USA), used to anchor the sequencing primers, and low-quality sequences were removed using the Trimmomatic software (Illumina, San Diego, CA, USA) [46]. Among the options available for running the Trimmomatic, SLIDINGWINDOW was used with the 4:20 option so that the end sequences were cut whenever the average quality was less than 30 (Phred Quality Score, Q ≥ 30), in four base intervals. The option MINLEN 74 was also used so that the sequences were eliminated if, after filtering, they had a length less than 74 bases, corresponding to the minimum expected size of the sequence. After trimming, the sequences were re-evaluated with FastQC to verify the filtering efficiency. Sequences trimmed in fastq format were converted to fasta format with the seqtk tool (Harvard University, Boston, MA, USA) [47] to evaluate frequencies in each run. Thus, it was possible to analyze the generated data with shell script, and to obtain a manageable and organized database of candidate aptamer sequences.

### 4.4. Evaluation of Specificity of the Selected Aptamers

Flow cytometry assays were performed with Caov-3, Ovcar-3, and Iose-144 cell lines. For each cell line, 0.5 × 10⁶ cells were incubated individually with 600 nM aptamers-FAM, then diluted in Binding Buffer containing 20% Fetal Bovine Serum for 1 h at 37 °C in an atmosphere with 5% of CO_2_. After the incubation time, cells were washed to remove the unbound aptamers. The median fluorescence intensity from the association of aptamers with cells was measured by FACSCanto II Cytometer (Plataforma de Citometria de Fluxo—Rede de Plataformas Tecnológicas Fiocruz, Rio de Janeiro, Brazil).

### 4.5. Three-Dimensional Structure Characterization of the Identified Aptamers

After identifying the sequences of the selected aptamers specific for OvCar-3 and CaoV3 cells, the five most abundant sequences present in the last round of each selection were modeled to characterize their structures. First, to improve the analysis and increase the reliability of the predictions, the 2D structures were obtained from two different servers: UNAfold Web Server (http://www.unafold.org/mfold/applications/dna-folding-form.php), accessed on 4 March 2022 and NUPACK: Nucleic Acid Package server (http://www.nupack.org/partition/new, accessed on 2 March 2022). Then, the DNA sequences were converted into RNA, and the three-dimensional structures were modeled using the RNAcomposer server (http://rnacomposer.cs.put.poznan.pl/, accessed on 4 March 2022) for each of the secondary structures obtained in the previous step. Then, the 3D RNA models were replaced into DNA models using the x3DNA server (http://web.x3dna.org/index.php/mutation, accessed on 4 March 2022), by the exchange of Uracils by Thymine. In addition, ribose sugar was replaced by deoxyribose by modifying the output pdb file from the x3DNA server after the exchange of nitrogenous bases. Finally, the structures were refined through geometry minimization using the software Phenix (Phenix Industrial Consortium, Berkeley, CA, USA) [48].

### 4.6. Identification of Target Proteins Potentially Recognized by Tumor-Specific Aptamers

Proteins which could be the potential targets for aptamers binding were selected from proteomic data from ovarian tumor cell lines provided by Coscia et al., 2016 and Faça et al., 2008. Proteins with upregulated expression in the tumor lines OvCar-3 and Caov-3 as compared to non-tumor ovarian control lines were selected. Proteins overexpressed at least 1.5× in tumor lines and without expression in control lines were considered; then, they were ranked according to expression levels in the respective cell lines. Subsequently, the selected proteins were evaluated for subcellular localization using the information available on the UniProt website (https://www.uniprot.org, accessed on 28 September 2022) and the PSORT WWW server (https://psort.hgc.jp/form2.html, accessed on 28 September 2022) [16,17].

### 4.7. Three-Dimensional Structure of the Selected Target Molecules

The three-dimensional structures of the proteins selected in the previous step were obtained from the PDB (Protein Data Bank) database [49], being considered the structures with the best coverage and experimental resolution. Proteins that did not have a three-dimensional structure experimentally characterized and deposited in the PDB were modeled using the AlphaFold Protein Structure database (https://alphafold.ebi.ac.uk, accessed on 5 August 2022), which is an AI system developed by DeepMind, capable of predicting a protein’s structure from its amino acid sequence [50].

### 4.8. Identification of Transmembrane Domains in the Selected Proteins

Proteins described as transmembrane had their transmembrane domains predicted through the ALOM [51], which identifies transmembrane segments, and MTOP [52], which predicts membrane topology, algorithms, available in PSORT II (https://psort.hgc.jp/form2.html, accessed on 15 October 2022) [53]. In addition, the OPM database (https://opm.phar.umich.edu, accessed on 15 October 2022) provided information about the spatial orientation of proteins in the lipid bilayer through the PPM 2.0 webserver (https://opm.phar.umich.edu/ppm_server2, accessed on 15 October 2022) [54].

### 4.9. Evaluation of the Electrostatic Potential of Selected Proteins

Proteins with transmembrane regions were evaluated for their atomic charge in order to determine the electrostatic potential of the molecular surface. The electrostatic potential analysis was conducted with the APBS program (https://server.poissonboltzmann.org) [55]. The amber force field charge and radii parameters were assigned using the PDB2PQR server, considering the pH of 7.5. The protein’s electrostatic potential was represented and colored as red, which indicated a negative charge, and blue, which indicated a positive charge.

### 4.10. Molecular Docking

The coordinates files of possible target proteins were pre-processing calculating pK values of ionizable groups and adding missing hydrogen atoms by web server H++ (http://newbiophysics.cs.vt.edu/H++/index.php, accessed on 15 October 2022) [56]. Calculations were performed using the following parameters: salinity = 0.15, internal dielectric = 10, external dielectric = 80, and pH7.5. Hydrogen atoms were also added to the aptamer structures by the Chimera software. The webserver HADDOCK 2.4 (https://wenmr.science.uu.nl/haddock2.4) was used for receptor–ligand interaction analysis. HADDOCK is an approach that consists of both rigid-body docking and semi-flexible refinement, and final refinement in explicit solvent. Dockings were run on server easy interfaces with default parameters. The constraints used in HADDOCK are Ambiguous Interaction Restraints (AIR). The residues with relative solvent accessibility >40% according to GETAREA [57] were defined as active residues. For ssDNA aptamers, the whole molecule was used as target, so all residues were defined as active residues. The HADDOCK score function is a weighted sum of energetic terms such as intermolecular electrostatic, van der Waals, desolvation and AIR energies. The best complex receptor–ligand was selected by lowest HADDOCK score and cluster size.

### 4.11. Analysis of Differentially Expressed Gene in Ovarian Cancer

The expression profile of FXYD3, ALPP, and TSPAN15 in ovarian cancer was analyzed in the TNMplot (https://www.tnmplot.com/, accessed on 15 October 2022), which is an available web platform that shows differential gene expression in normal, primary site tumor, and metastatic ovary cancer samples [58]. TNMplot includes normal (n = 46), tumor (n = 744), and metastatic (n = 44) ovarian cancer samples. STRING is a publicly accessible platform (https:string-dg.org, 15 October 2022) which allows the analysis of protein–protein interactions, and the TSPAN15, ALPP, and FXYD3 pathways constituents were analyzed.

### 4.12. Molecular Dynamics

Molecular dynamics (MD) simulations were conducted using different protocols for the aptamer and protein–aptamer complexes.

For aptamer systems, MD simulations were carried out using Amber 20 [59,60], DNA interactions were represented using amber BSC1 force-field. Electrostatic interactions were treated using the Particle-Mesh Ewald (PME) algorithm, with a cutoff of 12 Å. Each system was simulated under periodic boundary conditions in a triclinic box whose dimensions were automatically defined, considering 10 Å from the outermost DNA atoms in all Cartesian directions. The simulation box was filled with TIP3P water molecules. Subsequently, a two-step energy minimization procedure was performed: (i) 2000 steps (1000 steepest descent + 1000 conjugate-gradient) with all heavy atoms harmonically restrained with a force constant of 5 kcal mol-1 Å-2, and (ii) 5000 steps (2500 steepest descent + 2500 conjugate-gradient) without position restraints. Next, initial atomic velocities were assigned using the Maxwell–Boltzmann distribution, corresponding to a temperature of 20 K. The systems were gradually heated to 300 K over one nanosecond using the Langevin thermostat. All heavy atoms were harmonically restrained during this stage with a force constant of 10 kcal mol-1 Å-2. All systems were subsequently equilibrated during nine successive 500 ps equilibration simulations where position restraints approached zero progressively. After this period, all of the systems were simulated with no restraints at 300 K in the Gibbs ensemble with a 1 atm pressure using isotropic coupling. All chemical bonds containing hydrogen atoms were restricted using the SHAKE algorithm, and the time step was set to 2 fs. For each system, we simulated an MD run of 500 ns.

For each protein–aptamer complex, its orientation concerning the membrane was determined by the Positioning of Proteins in Membrane (PPM) server of the Orientations of Proteins and Membranes (OPM) database [54]. The oriented structures were embedded in a fully hydrated POPC (1-palmitoyl-2-oleoyl-sn-glycero-3-phosphocholine) lipid bilayer using the CHARMM-GUI Membrane builder (Lehigh University, Bethlehem, PA, USA) [61]. The complex systems were then hydrated using the TIP3 water model and 0.15 M KCl ion concentration. MD simulations were carried out using Amber 20 [59,60]. Protein and DNA interactions were represented using amber ff14SB and BSC1 force-field, respectively, and amber Lipid14 force-field applied to lipids. Electrostatic interactions were treated using the Particle-Mesh Ewald (PME) algorithm, with a cutoff of 10 Å. Subsequently, a two-step energy minimization procedure was performed: (i) 2000 steps (1000 steepest descent + 1000 conjugate-gradient) with all heavy atoms harmonically restrained with a force constant of 5 kcal mol-1 Å-2; (ii) 5000 steps (2500 steepest descent + 2500 conjugate-gradient) without position restraints. Next, initial atomic velocities were assigned using a Maxwell–Boltzmann distribution corresponding to an initial temperature of 20 K, and the systems were gradually heated to 300 K over one nanosecond using the Langevin thermostat. During this stage, all heavy atoms were harmonically restrained with a force constant of 10 kcal mol-1 Å-2. Systems were subsequently equilibrated using a sequence of two steps. In the first step, only the heavy atoms from the lipids were equilibrated during nine successive 500 ps equilibration simulations, where position restraints approached zero progressively. The heavy atoms restraints were removed for the protein–aptamer complex in step two, during nine successive 500 ps equilibrations. After this period, all the systems were simulated with no restraints at 300 K in the Gibbs ensemble, with a 1 atm pressure using a semi-isotropic coupling. We simulated a 300 ns MD run for the AptaC2–FXYD3 system, and a 100 ns MD run for the AptaO1–TSPAN15 and AptaO2–TSPAN15 systems.

Simulation trajectories were analyzed with GROMACS package tools (GROMACS Development Team, Uppsala, Sweden) [62]. Root-mean-square deviation (RMSD) values and Radius of Gyration (Rg) were calculated separately for each system fitting their heavy atoms, taking the initial structure of the production dynamics as a reference. Conformational clusterization for an aptamer was performed using the GROMOS method with a cutoff of 5.0 Å, considering all atoms. The central structure of the largest cluster was taken for additional analysis. In addition, dynamic secondary structure along the trajectories was assessed using the Barnaba [63]. Hydrogen bonds (H-bond) were calculated between protein and aptamer complexes. H-bond formation was defined using a geometric criterion with CPPTRAJ [64] in Amber (Amber Software, Paris, France). We considered a hit when the distance between two polar heavy atoms, with at least one hydrogen atom attached, was less than 3.5 Å and using an H-donor angle higher than 120°. The binding free energy decomposition of protein–aptamer complexes was calculated by extracting the uncorrelated 500 snapshots from each MD simulation’s last 50 ns trajectory, using the MM/GBSA (molecular mechanics generalized Born surface area) approach. The interaction energy and solvation free energy for the complex, receptor, ligand, and resulting averages were calculated using the MMPBSA.py module [65], available in the AMBER distribution.

### 4.13. Tissue Microarray

Ovarian disease spectrum tissue microarray slides (US Biomax—ref.: OV1005b) were kept at 60° for 2 h before use. For the deparaffinization step, the slides were bathed in xylol (3× for 5min), then the sections were dehydrated in decreasing concentrations of ethanol (100%, 95%, 80% and 70%—5 min each). Then, tissues were hydrated in water for 5 min. For antigenic recovery, the slides were placed in a streamer containing Citrate pH 6.0/Tris EDTA pH 9.0/Trilogy™ (Cell Marque, Rocklin, CA, USA) solution for 30 min. Then, after washes with TBS (3× for 5min), the nonspecific binding was blocked with Novolink™ Protein Block (Leica Biosystems, Deer Park, IL, USA) for 5 min. The slides were incubated with aptamers-FAM, AptaC2, and AptaO1, at a concentration of 400nM for 1 h at room temperature, then washed with TBS and incubated with DAPI (1:5000 for 10 min). After washing in TBS, ProLong™ Gold Antifade (Thermo Fisher, Waltham, MA, USA) was used for mounting the slides.

## 5. Patents

The Aptamer sequences are protected under patent filing PCT/BR2022/050356.

## Figures and Tables

**Figure 1 ijms-24-06315-f001:**
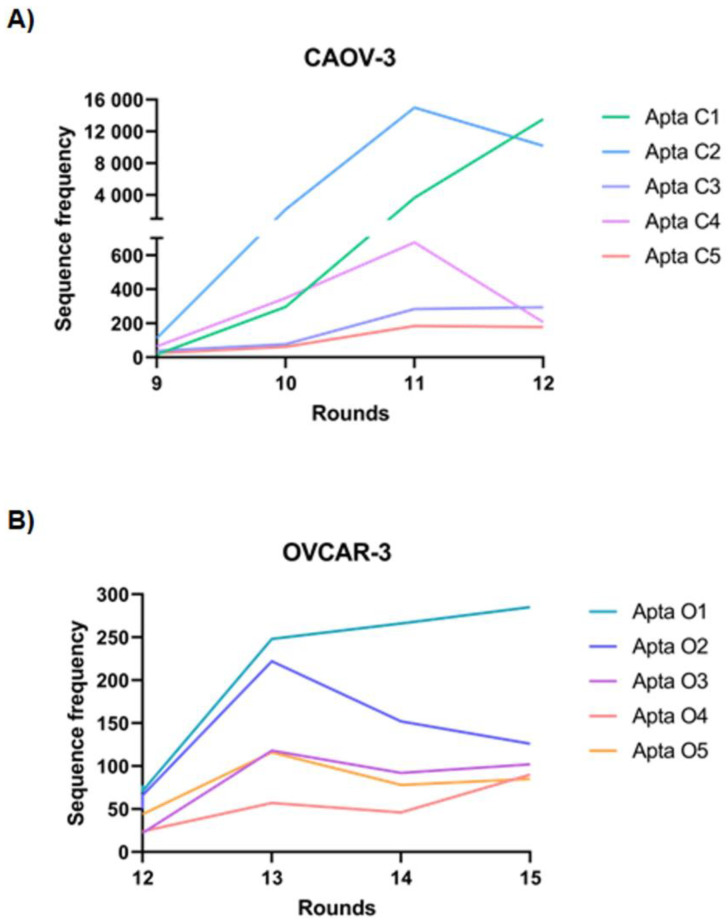
Graph demonstrating the enrichment of the five most abundant individual aptamers in the last rounds of Cell-Selex. (**A**) Aptamers frequency selected for Caov-3 cell line: AptaC1, AptaC2, AptaC3, AptaC4, and AptaC5 over the R9, R10, R11, and R12 rounds. (**B**) Aptamers frequency selected for OvCar-3 cell line: AptaO1, AptaO2, AptaO3, AptaO4, and AptaO5 over the R12, R13, R14, and R15 rounds.

**Figure 2 ijms-24-06315-f002:**
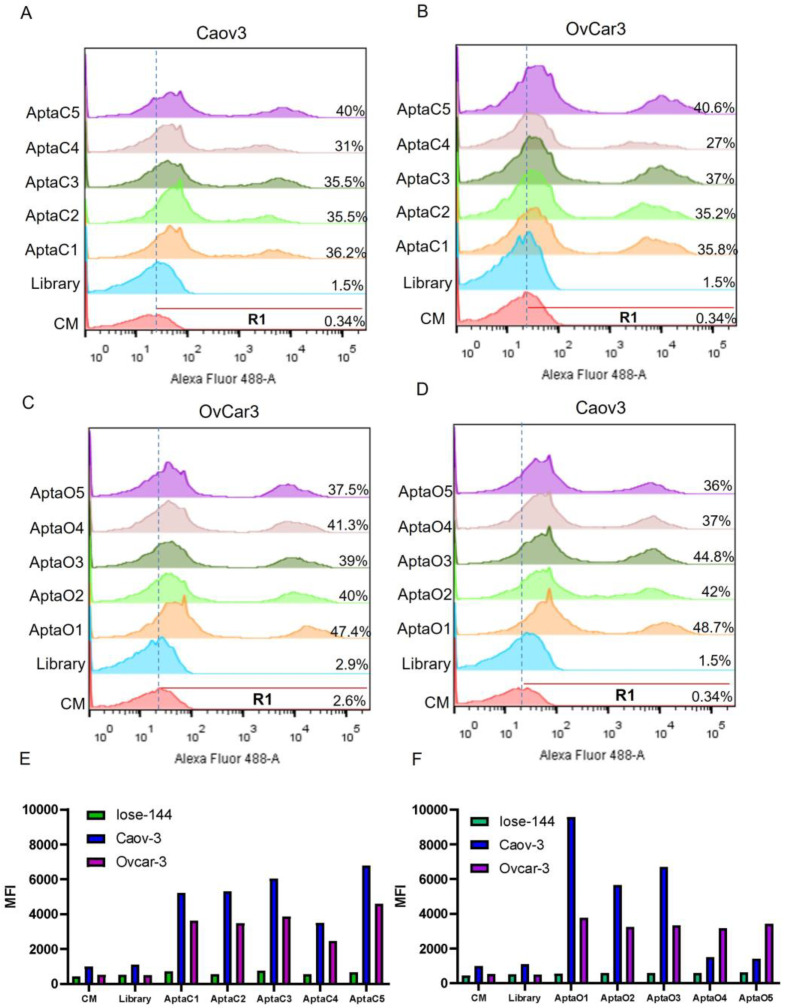
Specificity evaluation of the five most abundant individual aptamers in the last round. Labeling Intensity of the five individual aptamers selected for Caov-3 AptaC1, AptaC2, AptaC3, AptaC4, and AptaC5 in: (**A**) Caov-3 cells, and (**B**) Ovcar-3 cells. Labeling Intensity of the five individual aptamers selected for Ovcar-3: AptaO1, AptaO2, AptaO3, AptaO4, and AptaO5 in (**C**) Ovcar-3 cells and (**D**) Caov-3 cells. Graph demonstrating the Median Fluorescence Intensity (MFI) of the five individual aptamers selected for (**E**) Caov-3 and (**F**) Ovcar-3 in Iose-144 non-tumor control cells and Caov-3 and Ovcar-3 tumor cells. Negative controls correspond to culture medium (CM) and unbound unspecific aptamers from the first round of selection (Library).

**Figure 3 ijms-24-06315-f003:**
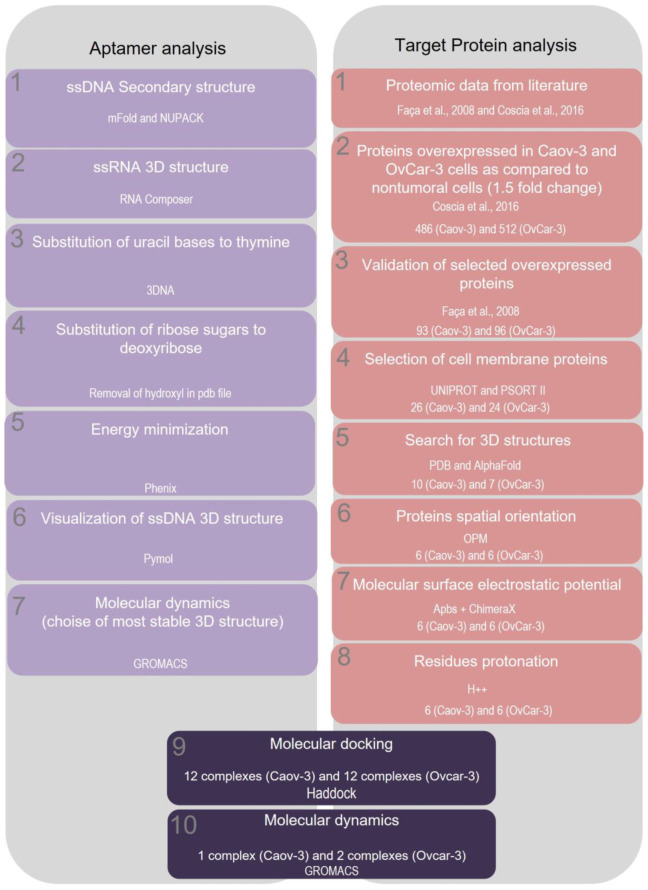
Workflow demonstrating the sequential steps for aptamer analysis and potential protein target analysis. Proteomic data was obtained by Coscia et al., 2016 and Faça et al., 2008 [16,17].

**Figure 4 ijms-24-06315-f004:**
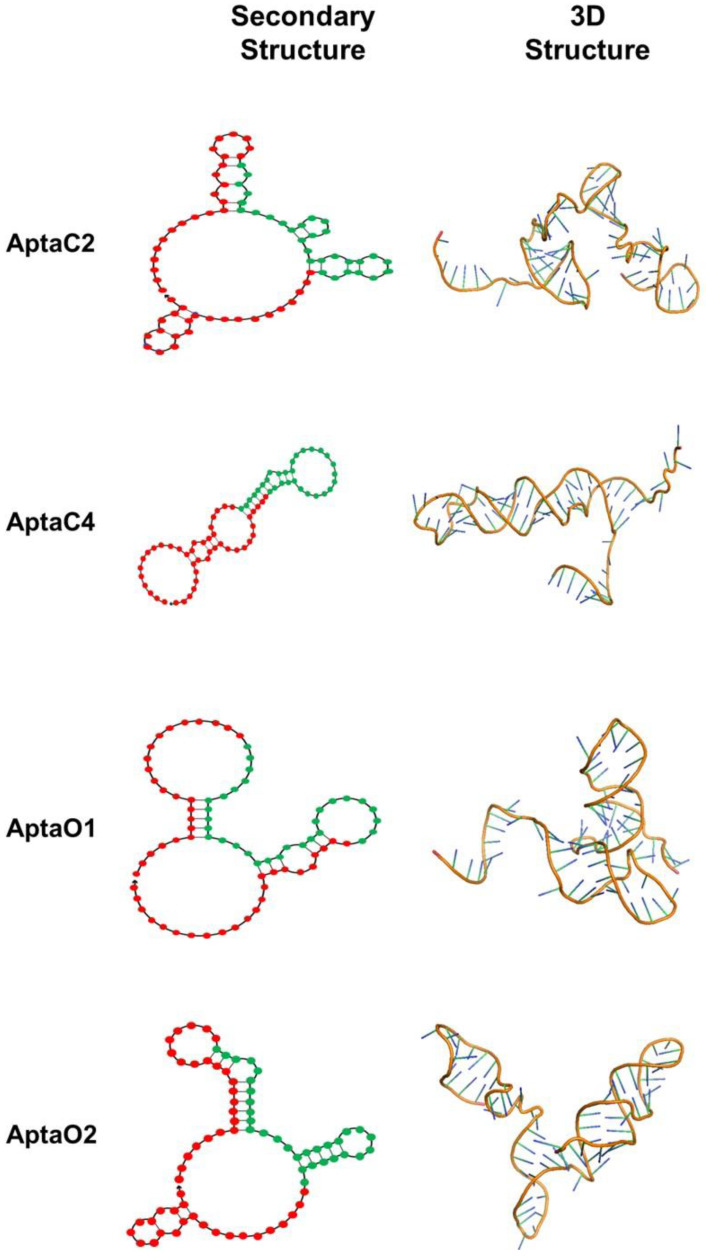
Characterization of secondary and tertiary aptamer structures. The aptamers from Caov-3 (AptaC2 and AptaC4) and from OvCar-3 (AptaO1 and AptaO2), presenting the lower ΔG, were described. In the secondary structures, the nucleotides marked in red correspond to the adapter sequences, while the nucleotides marked in green correspond to the central core N30.

**Figure 5 ijms-24-06315-f005:**
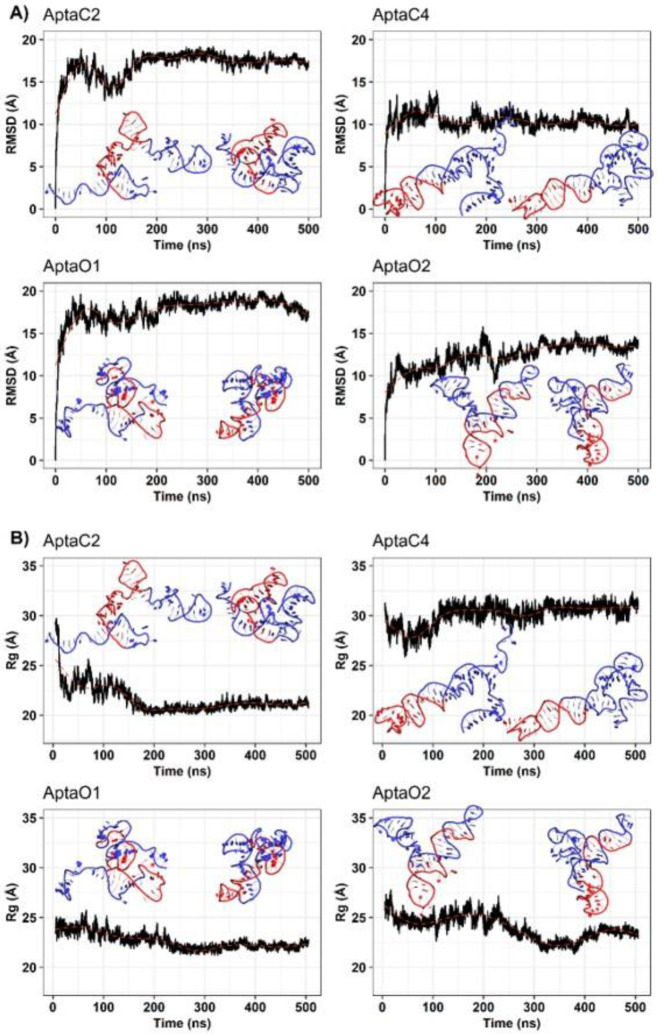
Root-mean-square deviations (**A**) and radius of gyration (**B**) throughout the molecular simulations. Aptamer structure representations corresponding to the initial production structure and the central structure resulting from clustering analysis of simulation are displayed within each plot. The regions of the adapter and central core are depicted in blue and red, respectively.

**Figure 6 ijms-24-06315-f006:**
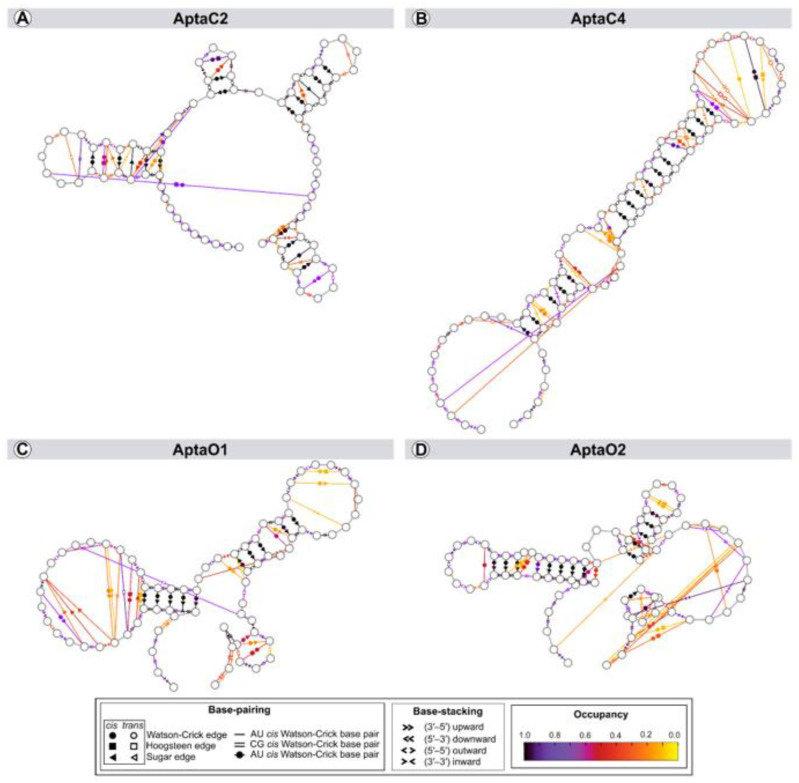
Dynamic secondary structure representation of aptamers AptaC2 (**A**); AptaC4 (**B**); AptaO1 (**C**); and AptaO2 (**D**). The extended secondary structure annotation follows the Leontis–Westhof classification (see legend caption). The color scheme shows the fraction of frames (occupancy) within the system for which the interaction was formed.

**Figure 7 ijms-24-06315-f007:**
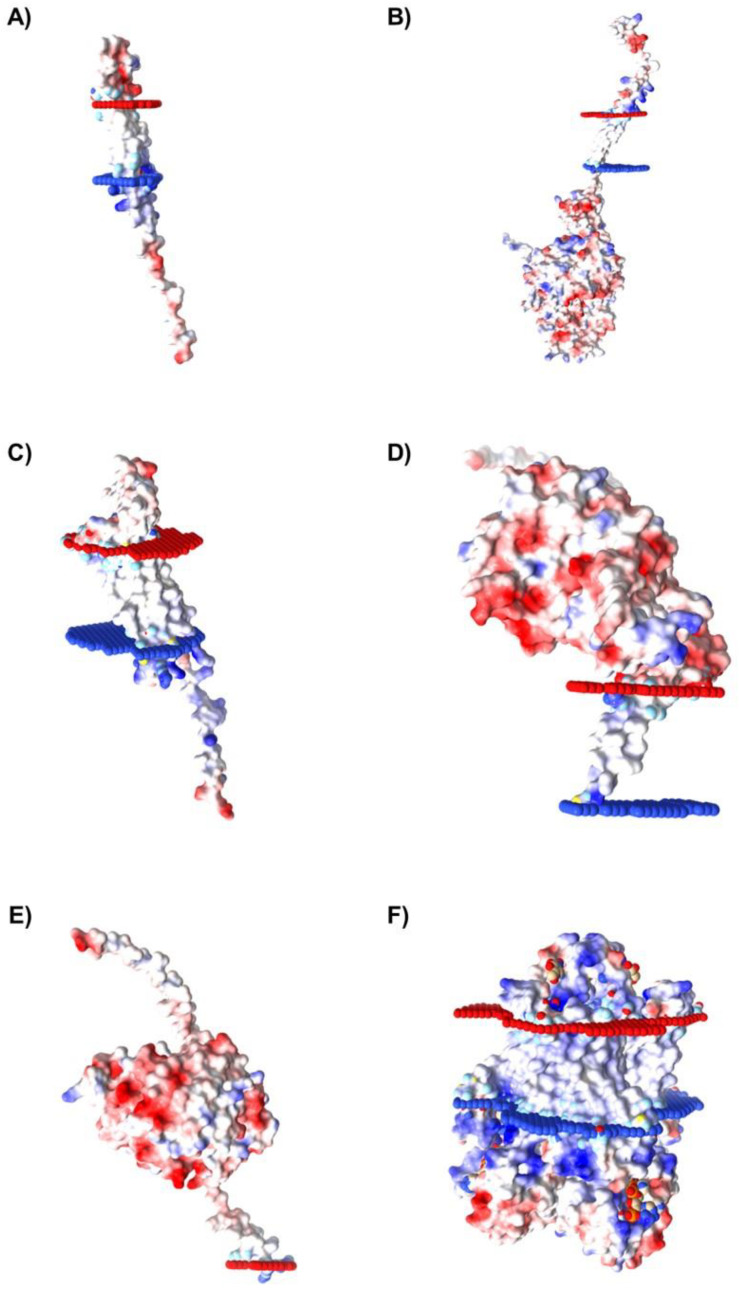
Membrane regions’ spatial distribution and electrostatic potential determination of proteins identified for Caov-3 cell line. The proteins selected for Caov-3 cells were: (**A**) FXYD3, (**B**) ITGB2, (**C**) CLDN7, (**D**) ALPP, (**E**) SMPDL3B, and (**F**) STEAP4. For the spatial orientation, the red portion indicates the extracellular region, and the blue portion indicates the intracellular region. For the electrostatic potential determination, blue color indicates a positive charge and red color indicates a negative charge.

**Figure 8 ijms-24-06315-f008:**
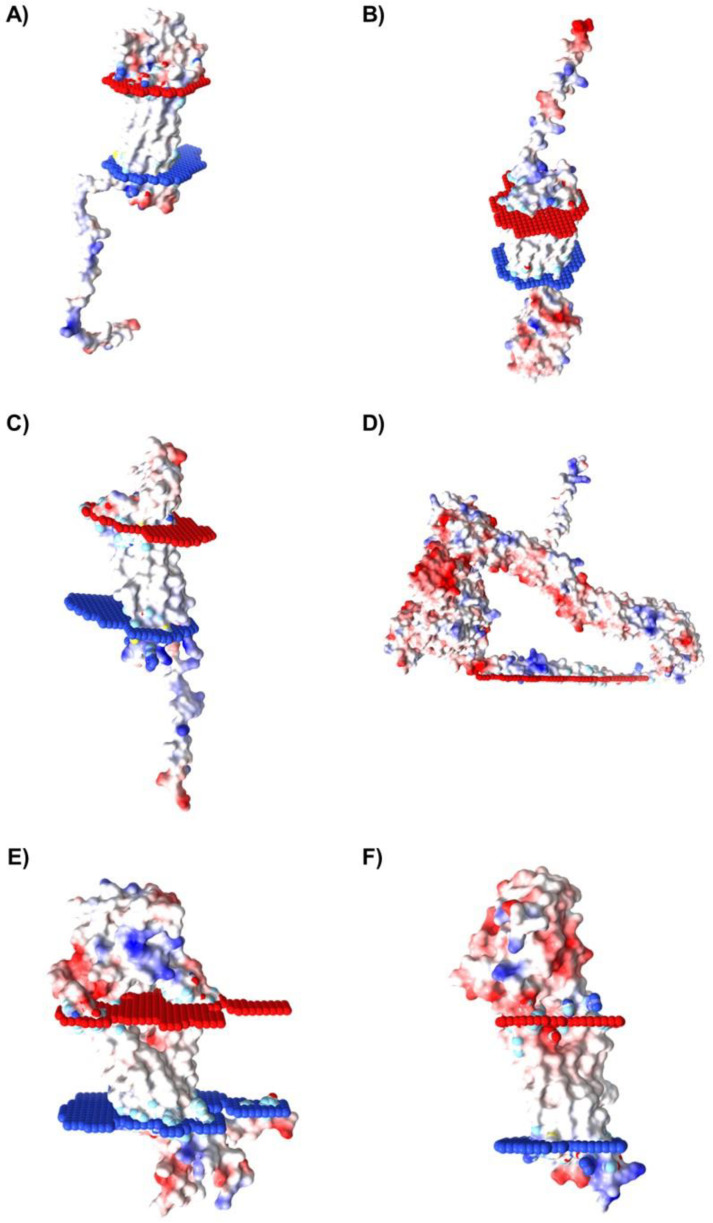
Membrane regions spatial distribution and electrostatic potential of proteins identified for OvCar-3 cell line. The proteins selected for OvCar-3 cells were: (**A**) CLDN6, (**B**) CD47, (**C**) CLDN7, (**D**) EPHA1, (**E**) TSPAN15, and (**F**) UPK1B. For the spatial orientation, the red portion indicates the extracellular region, and the blue portion indicates the intracellular region. For the electrostatic potential evaluation, blue color indicates a positive charge and red color indicates a negative charge.

**Figure 9 ijms-24-06315-f009:**
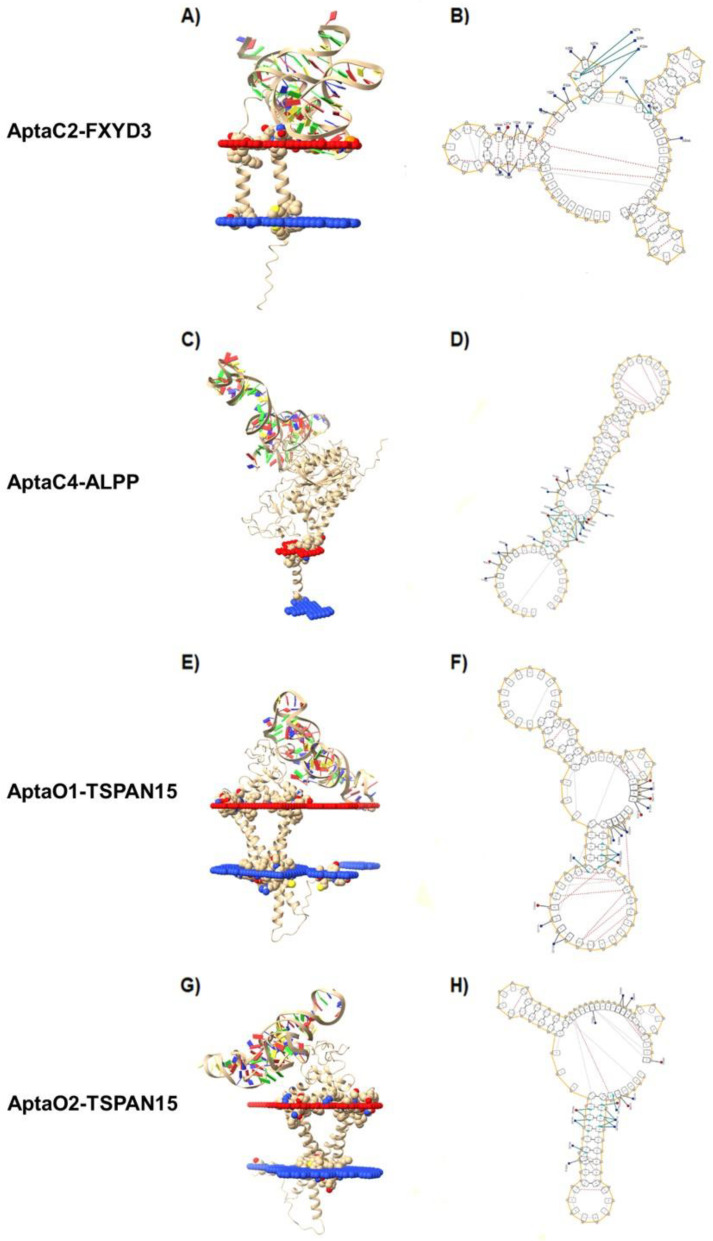
The best binding complexes for potential protein targets in Caov-3 and OvCar-3 cell lines. Image representing the best clusters from molecular docking for: (**A**) AptaC2-FXYD3, (**C**) AptaC4-ALPP, (**E**) AptaO1-TSPAN15, and (**G**) AptaO2-TSPAN15. Residues and interactions between the binding complexes: (**B**) AptaC2-FXYD3, (**D**) AptaC4-ALPP, (**F**) AptaO1-TSPAN15, and (**H**) AptaO2-TSPAN15.

**Figure 10 ijms-24-06315-f010:**
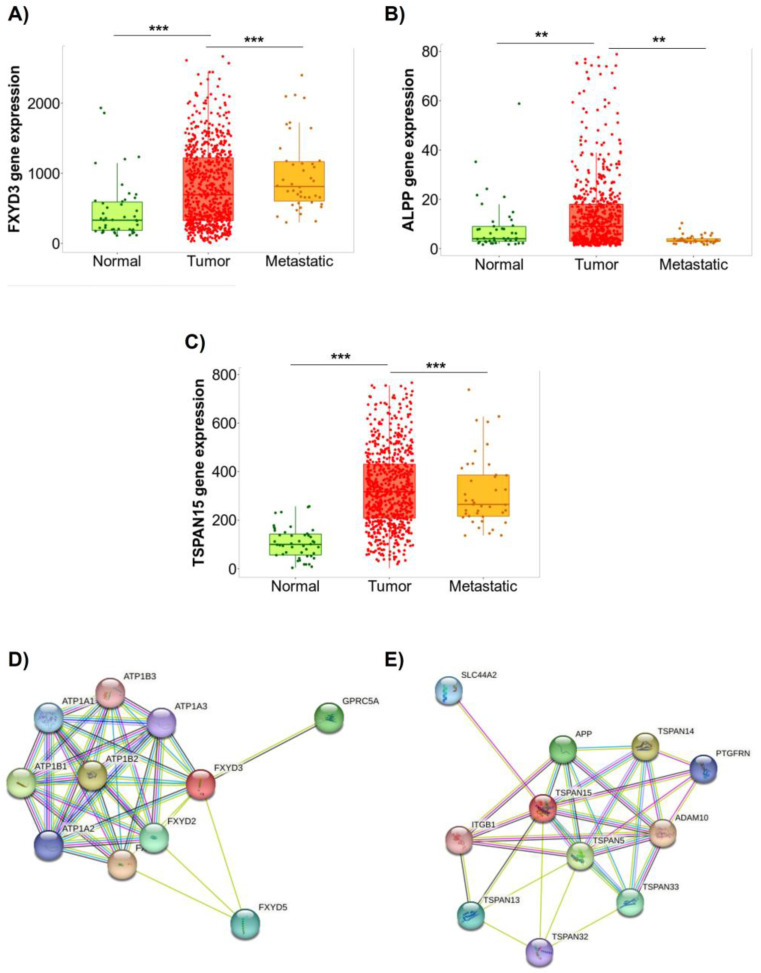
Gene expression profile of the selected targets in ovarian cancer and PPI network analysis. Samples from ovarian cancer patients comparing normal, primary site tumor, and metastatic tissues from genetic chip data available in the TNMplot database for (**A**) FXYD3, (**B**) ALPP, and (**C**) TSPAN15. The y axis represents mRNA expression, and the x axis represents the studied groups. The main signaling pathways were triggered by (**D**) FXYD3 and (**E**) TSPAN15 in ovarian cancer by STRING database. (**) indicates *p* value < 0.05. (***) indicates *p* value < 0.001.

**Figure 11 ijms-24-06315-f011:**
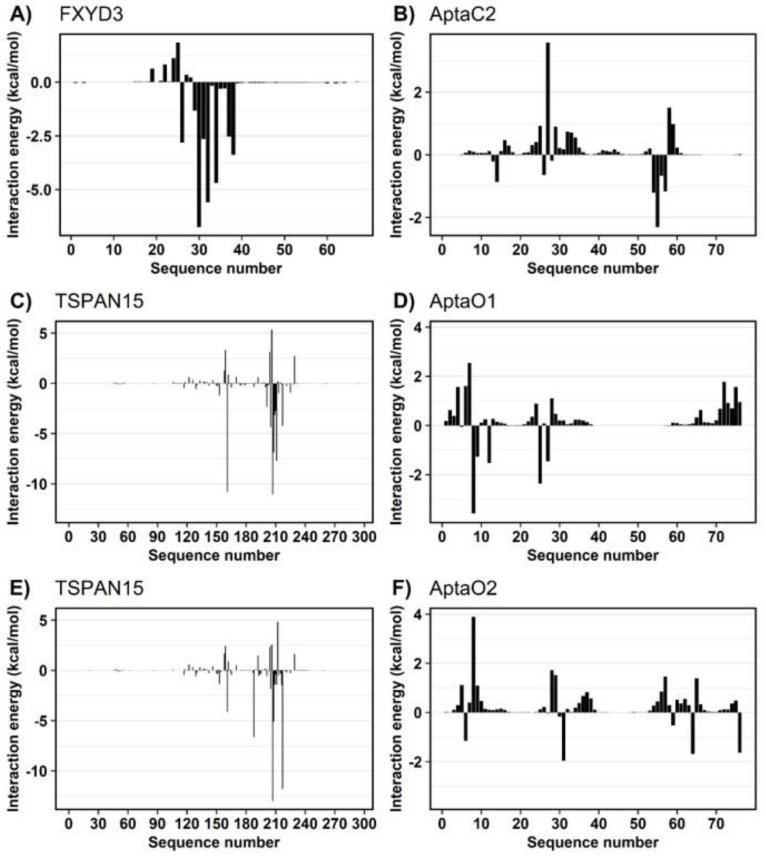
Decomposition of the binding free energy for the three simulated systems.

**Figure 12 ijms-24-06315-f012:**
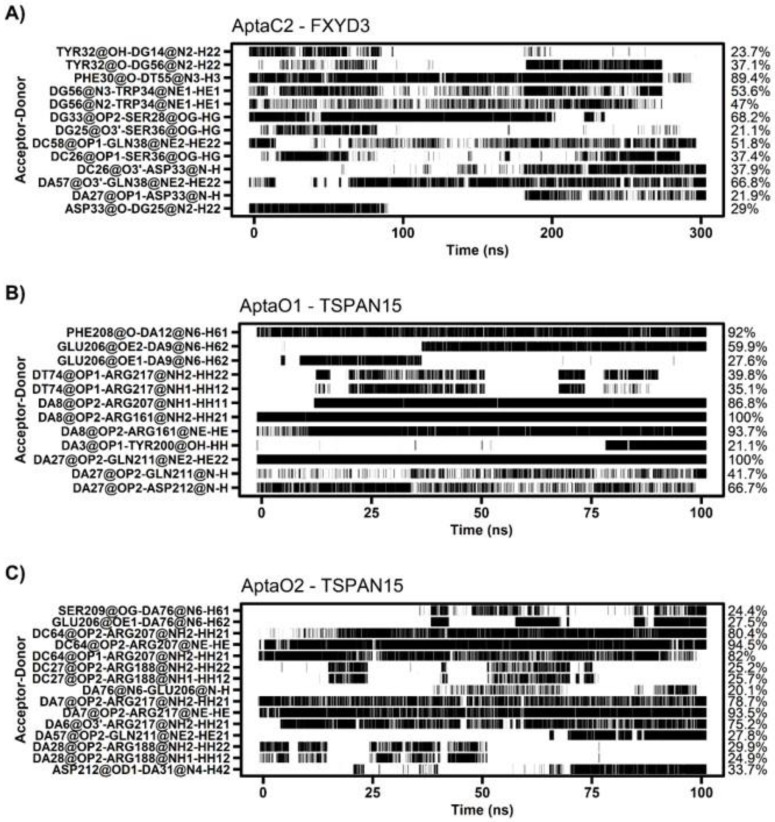
Hydrogen bond between protein and aptamer of each system. (**A**) AptaC2-FXYD3; (**B**) AptaO1-TSPAN15; (**C**) AptaO2-TSPAN15. The occupancy of each interaction is on the right axis.

**Figure 13 ijms-24-06315-f013:**
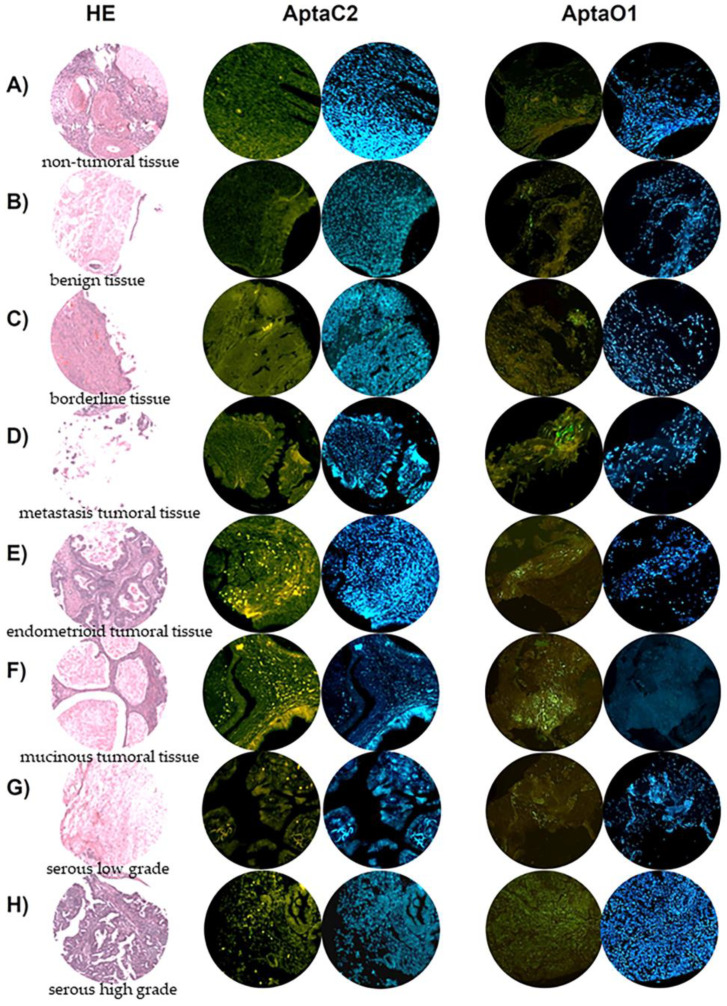
Tissue microarray analysis for the evaluation of AptaC2 and AptaO1 detection capacity for different ovarian tumor samples. Images demonstrate different stages and histologic subtypes of ovarian cancer: (**A**) non-tumoral tissue, (**B**) benign tissue, (**C**) borderline tissue, (**D**) metastasis tumoral tissue, (**E**) subtype endometrioid tumoral tissue, (**F**) subtype mucinous tumoral tissue, (**G**) subtype serous low grade, and (**H**) serous high grade, marked with: HE, AptaO1-FAM, and DAPI.

**Table 1 ijms-24-06315-t001:** Molecular docking parameters of possible targets in Caov-3 cell-line.

	AptaC2				AptaC4			
	Cluster	HADDOCK Score	Cluster Size	RMSD	Cluster	HADDOCK Score	Cluster Size	RMSD
**ALPP**	Cluster 5	−21.9 +/− 30.2	5	0.7 +/− 0.4	**Cluster 3**	**−27.6 +/− 9.6**	**7**	**1.0 +/− 0.9**
**CLDN7**	Cluster 2	−2.7 +/− 17.9	20	0.8 +/− 0.5	Cluster 11	25.7 +/− 12.5	4	23.6 +/− 0.5
**FXYD3**	**Cluster 3**	**−40.7 +/− 10.5**	**11**	**1.4 +/− 0.8**	Cluster 6	33.9 +/− 12.1	12	1.4 +/− 0.9
**ITGB2**	Cluster 4	−24.9 +/− 2.6	17	10.9 +/− 0.2	Cluster 5	21.8 +/− 6.3	9	22.9 +/− 0.2
**SMPDL3B**	Cluster 12	−18.8 +/− 7.3	5	18.6 +/− 0.1	Cluster 4	15.0 +/− 5.5	11	13.2 +/− 0.8
**STEAP4**	Cluster 9	−49.4 +/− 20.1	7	9.4 +/− 0.1	Cluster 4	−10.8 +/− 9.3	12	2.2 +/− 1.8

**Table 2 ijms-24-06315-t002:** Molecular docking parameters of possible targets in OvCar-3 cell-line.

	AptaO1				AptaO2			
	Cluster	HADDOCK Score	Cluster Size	RMSD	Cluster	HADDOCK Score	Cluster Size	RMSD
**CD47**	Cluster 10	5.2 +/− 12.8	8	5.5 +/− 0.1	Cluster 11	−8.9 +/− 20.8	5	3.2 +/− 2.3
**CLDN6**	Cluster 9	4.3 +/− 2.9	6	19.2 +/− 0.3	Cluster 2	14.9 +/− 5.7	12	15.6 +/− 0.2
**CLDN7**	Cluster 1	2.3 +/− 9.7	13	14.9 +/− 0.2	Cluster 1	3.6 +/− 9.2	14	1.5 +/− 1.1
**EPHA1**	Cluster 1	8.5 +/− 4.1	26	5.9 +/− 0.1	Cluster 4	18.9 +/− 3.4	9	15.5 +/− 0.2
**TSPAN15**	**Cluster 4**	**−36.8 +/− 6.9**	**13**	**13.9 +/− 0.3**	**Cluster 1**	**−11.5 +/− 17.7**	**26**	**25.5 +/− 0.1**
**UPK1B**	Cluster 11	6.0 +/− 17.4	6	1.1 +/− 0.9	Cluster 2	19.1 +/− 7.1	14	2.1 +/− 1.2

**Table 3 ijms-24-06315-t003:** Binding free energies for complexes calculated by the MM/GBSA method.

Items	AptaC2—FXYD3	AptaO1—TSPAN15	AptaO2—TSPAN15
ΔE_vdw_	−75.93 ± 0.31	−80.99 ± 0.53	−80.46 ± 0.41
ΔE_ele_	1579.04 ± 6.56	1627.33 ± 11.74	1498.08 ± 7.76
ΔG_egb_	−1510.61 ± 6.32	−1563.69 ± 11.16	−1422.75 ± 7.51
ΔG_esurf_	−8.36 ± 0.03	−10.03 ± 0.06	−10.13 ± 0.04
ΔG_ele+egb_ ^a^	68.43 ± 6.44	63.64 ± 11.45	75.33 ± 7.63
ΔG_bind_ ^b^	−15.86 ± 0.39	−27.39 ± 0.50	−15.27 ± 0.48

ll mean and standard error values are given in kcal/mol. ^a^ ΔG_ele+egb_ = ΔE_ele_ + ΔG_egb._
^b^ ΔG = ΔE_vdw_ + ΔE_ele_ + ΔG_esurf_ + ΔG_egb._

## Data Availability

Data supporting reported results can be found in DOI:10.5281/zenodo.7221846.

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
