# Peer review of "Next Generation of Ovarian Cancer Detection Using Aptamers"

_ijms, 2023, doi:10.3390/ijms24076315_

Round 1

Reviewer 1 Report

The authors introduced interesting methods to find specific aptamers to improve tumor diagnosis in ovarian cancer. In addition, the manuscript demonstrated that various computer modeling has been very helpful for aptamer screening and target molecule identification. The structure of manuscript was well organized, and the English was also well written.  

1. Despite the advantages of aptamers for the diagnosis of tumors, they are well known for their strong background noise signals. So, researchers try to show the superiority of aptamers by comparing specific antibodies and aptamers. Have you ever done any comparisons?

2. It has been reported that some aptamers cause cell cytotoxicity. Have you ever tested toxicity with the aptamers developed in this study?

Reviewer 2 Report

In this manuscript, da Silva Abreu and colleagues select aptamers using Cell-SELEX in two OVCA cell lines and analyze these aptamers and their potential targets in ovarian cancer cells. The authors narrow down 4 aptamers C2, C4, O1, and O2 and using MD, identify FXYD3, ALPP, and TSPAN15, as potential for each aptamer (TSPAN15 for O1/O2).

Authors should validate FXYD3 as a target for C2, ALPP as a target for C4, and TSPAN15 as a target for O1 and O2 in cancer cells. This can be done relatively easily. For instance, one can use si/shRNA to knockdown these proteins and test the efficacy of these aptamers. If this is beyond the scope of this manuscript, potential experiments that can be done to test these targets for each respective target should be proposed in the discussion. Limitations of aptamers and tools used should also warrants further discussion. 

Minor- In figure 13, authors should label tissue type in the figure itself.  
